# The structural basis of proton driven zinc transport by ZntB

Cornelius Gati[1], Artem Stetsenko [2], Dirk J. Slotboom[2], Sjors H.W. Scheres [1] & Albert Guskov [2]

Zinc is an essential microelement to sustain all forms of life. However, excess of zinc is toxic, therefore dedicated import, export and storage proteins for tight regulation of the zinc concentration have evolved. In Enterobacteriaceae, several membrane transporters are involved in zinc homeostasis and linked to virulence. ZntB has been proposed to play a role in the export of zinc, but the transport mechanism of ZntB is poorly understood and based only on experimental characterization of its distant homologue CorA magnesium channel. Here, we report the cryo-electron microscopy structure of full-length ZntB from *Escherichia coli* together with the results of isothermal titration calorimetry, and radio-ligand uptake and fluorescent transport assays on ZntB reconstituted into liposomes. Our results show that ZntB mediates $Zn^{2+}$ uptake, stimulated by a pH gradient across the membrane, using a transport mechanism that does not resemble the one proposed for homologous CorA channels.

[1] MRC Laboratory of Molecular Biology, Francis Crick Avenue, Cambridge Biomedical Campus, Cambridge CB2 0QH, UK. [2] Groningen Biomolecular Sciences and Biotechnology Institute, University of Groningen, Nijenborgh 4, 9747AG Groningen, The Netherlands. Cornelius Gati and Artem Stetsenko contributed equally to this work. Correspondence and requests for materials should be addressed to S.H.W.S. (email: scheres@mrc-lmb.cam.ac.uk) or to A.G. (email: a.guskov@rug.nl)

Zinc is one of the few 'essential-but-also-toxic' divalent cations required for the cell and is an important 'token coin' in host:pathogen interactions[1]: whenever host organisms try to sequester all available zinc at the host:pathogen interface to reduce the virulence of invading bacteria[2], the latter employ highly specific uptake systems to scavenge zinc[3]. Conversely, if the zinc concentration is elevated in hosts to oppress pathogens[4], the latter regulate their intracellular zinc concentration by scaling up the export of zinc[3]. Due to this ambidexterity, the tight regulation of zinc homeostasis is crucial. Different bacteria cope with this task in a variety of ways—for example, by storage of zinc by metallothioneins as in cyanobacteria[5], by assembly of redundant importers as in *Cupriavidus metallidurans*[6] or via a controlled shunt of zinc export–import as in *Escherichia coli*, where zinc–iron permeases (ZIPs) family transporter ZupT[7] and the ATP-binding cassette (ABC) transporter ZnuABC[8,9] are recruited for import, and P-type ATPase ZntA[10] and cation-diffusion facilitator YiiP[11] for export of zinc (Supplementary Fig. 1). In addition, the zinc transporter ZntB, which belongs to the CorA metal ion transporter (MIT) family is widespread in Enterobacteriaceae[12,13]. There is controversy over the question whether ZntB is an exporter[12] or importer[6]. Furthermore, mechanistic insight is lacking because crystal structures are available of only cytoplasmic parts of ZntB[14,15], and scarce transport activity measurements have been performed only in whole cells. We have obtained the structure of full-length ZntB from *E. coli* and performed isothermal titration calorimetry (ITC), radiolabelled zinc uptake and fluorescent transport experiments with ZntB reconstituted into liposomes. This study shows that ZntB mediates $Zn^{2+}$ transport, which is stimulated by a pH gradient across the membrane. The comparison of the full-length structure of ZntB with previously resolved structures of ZntB soluble domains in different conditions (in the presence and absence of $Zn^{2+}$) and structures of homologous CorA proteins, is indicative that ZntB and CorA proteins utilize different transport mechanisms.

## Results

**Structure of ZntB**. The *apo* structure of ZntB was obtained by single-particle cryo-electron microscopy (cryo-EM) using *n*-dodecyl-β-D-maltopyranoside (DDM)-solubilized and purified *E. coli* ZntB (EcZntB) (pre-treated with ethylenediaminetetraacetic acid (EDTA)) and resolved at an overall resolution of 4.2 Å (Supplementary Figs. 2 and 3, Table 1). The structure of ZntB revealed a pentameric arrangement (Fig. 1a, b), similar to

that reported for other members of the CorA family[13,16,17] (Supplementary Fig. 4) even though CorA and ZntB share very little sequence identity (below 20%) (Supplementary Fig. 5). Each protomer of ZntB consists of a large N-terminal cytoplasmic domain folded into an αβα motif. A long α-helix protrudes from the cytoplasmic domain into the membrane (TM1) and is joined to a second transmembrane helix (TM2) (Fig. 1a) via the only periplasmic loop, which bears the signature motif GxN of the CorA MIT family[13] (Fig. 1b, c). In the resolved structure of EcZntB, the cytoplasmic domain is very similar to the isolated domain of ZntB from *Vibrio parahaemolyticus* (Vp)[14] (rmsd ~2.5 Å), but significantly different from the homologous domain of *Salmonella typhimirium* (St) ZntB[15] (rmsd ~12 Å). Taking into account very high sequence conservation between EcZntB and StZntB of 92.6% (Supplementary Fig. 5), the structural difference is intriguing. This difference could be attributed to the two structures representing two different states in the transport cycle (discussed below). Analysis of the substrate translocation pore revealed a wider profile in ZntB than in the magnesium channels TmCorA and MjCorA (Supplementary Fig. 6). All three proteins have short extracellular loops between TM1 and TM2, where the family signature motif GxN that forms the selectivity filter (Fig. 1b, c) is located. Whereas in CorA proteins the signature motif has the sequence GMN, ZntBs show a Met to Val substitution, potentially having consequences for the substrate recognition, as in ZntB, the radius of the filter is ~4.5 vs 3.5–4.0 Å in CorAs (Fig. 1d).

**Substrate selectivity of ZntB**. One of the puzzling features of the CorA family is substrate selectivity. The geometry of the selectivity filter is thought to define the correct distances between (partially) hydrated cation and amino acid side chains of the filter, and hence recognition[18,19]. Intriguingly, although the hydrated radii of known transported substrates for the CorA family are very similar (2.10 Å, 2.09 Å, 2.10 Å, 2.07 Å for $Zn^{2+}$[20], $Mg^{2+}$[21], $Co^{2+}$[21,22] and $Ni^{2+}$[23], respectively) and all these cations have a similar octahedral arrangement of six water molecules in their first hydration shell in aqueous solution; different subfamilies have distinct substrate specificities. It has been proposed that this selectivity might be based on the rigidity of ion solvation shells and rates of water exchange[18].

To characterize the specificity, we performed ITC experiments and fluorescent transport assays. ITC experiments revealed binding of $Zn^{2+}$, $Cd^{2+}$, $Ni^{2+}$ and $Co^{2+}$ to ZntB, with $K_d$ values of 11.5, 22.6, 87.7 and 175.4 μM, respectively, in 1:1 stoichiometry

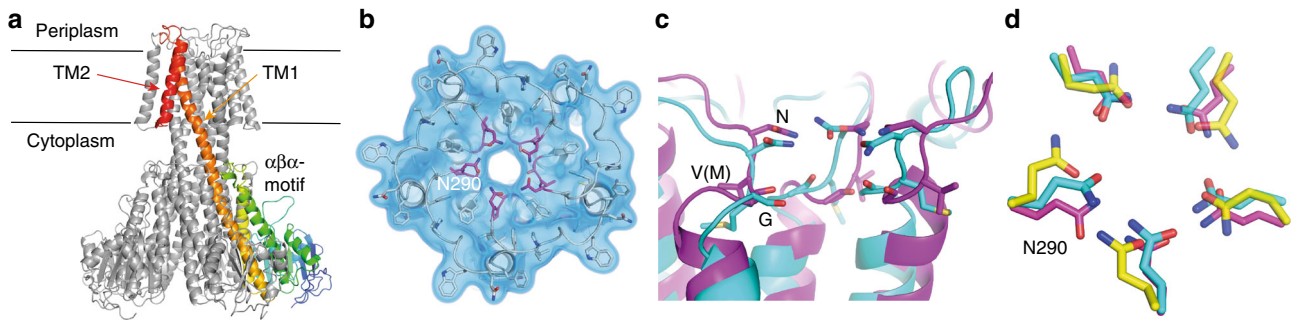

**Fig. 1** The structure of the full-length ZntB. **a** Side view, four subunits of ZntB pentamer are coloured grey, and one is coloured rainbow from blue (N-terminus) to red (C-terminus); the position of the membrane is indicated, trans membrane helices 1 and 2 as well as αβα-motif are labelled. **b** Top view (from periplasm) onto ZntB—10 trans membrane helices are arranged cylindrically, with TM2 ring at the periphery. Experimental density is contoured in blue. The connecting loops provide residues for the selectivity filter (in magenta), further exemplified in (**c**) structural comparison of selectivity filters from EcZntB (magenta) and TmCorA (cyan), only three out five monomers are shown (**d**). The overlay of Asn rings (of GxN motif) from ZntB (magenta), TmCorA (cyan) and MjCorA (yellow). Note that ZntB forms a slightly wider entry point to the pore

(Fig. 2). No binding of $Mn^{2+}$, $Mg^{2+}$ and $Cu^{2+}$ was detected. The ability of ZntB proteins to select $Zn^{2+}$ over $Mg^{2+}$, and conversely the ability of CorA proteins to select $Mg^{2+}$ over $Zn^{2+}$, may be explained by different sizes of the pore—the substitution of a single amino acid in the signature motif (see above) might be enough. However, $Co^{2+}$ and $Ni^{2+}$ appear to be substrates of both ZntB and CorA[24,25], thus the precise determinants of selectivity are still elusive. We additionally tested the transport of the cations

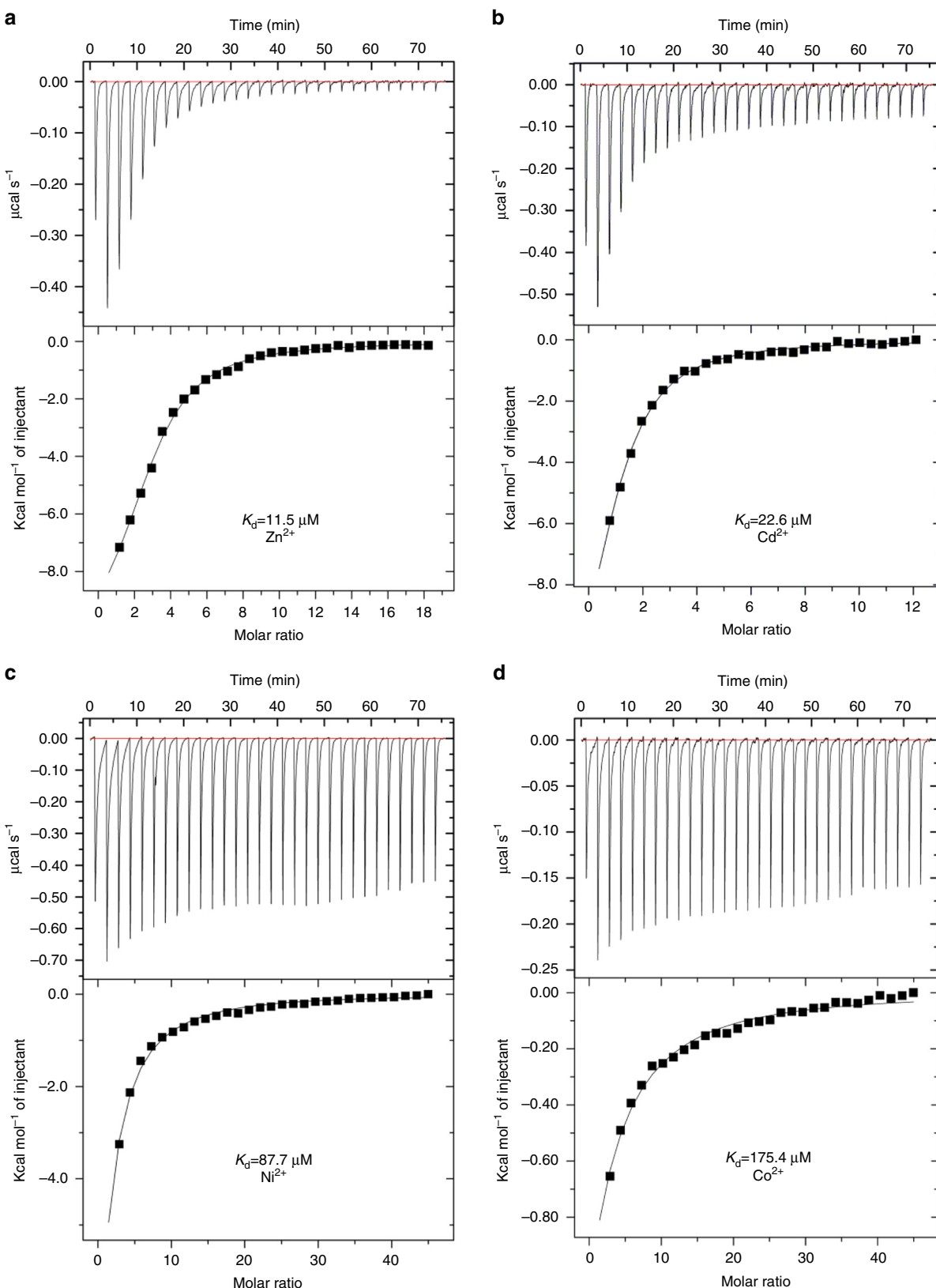

**Fig. 2** ITC profiles of substrate binding to ZntB of (**a**) $Zn^{2+}$ (**b**) $Cd^{2+}$ (**c**) $Ni^{2+}$ (**d**) $Co^{2+}$

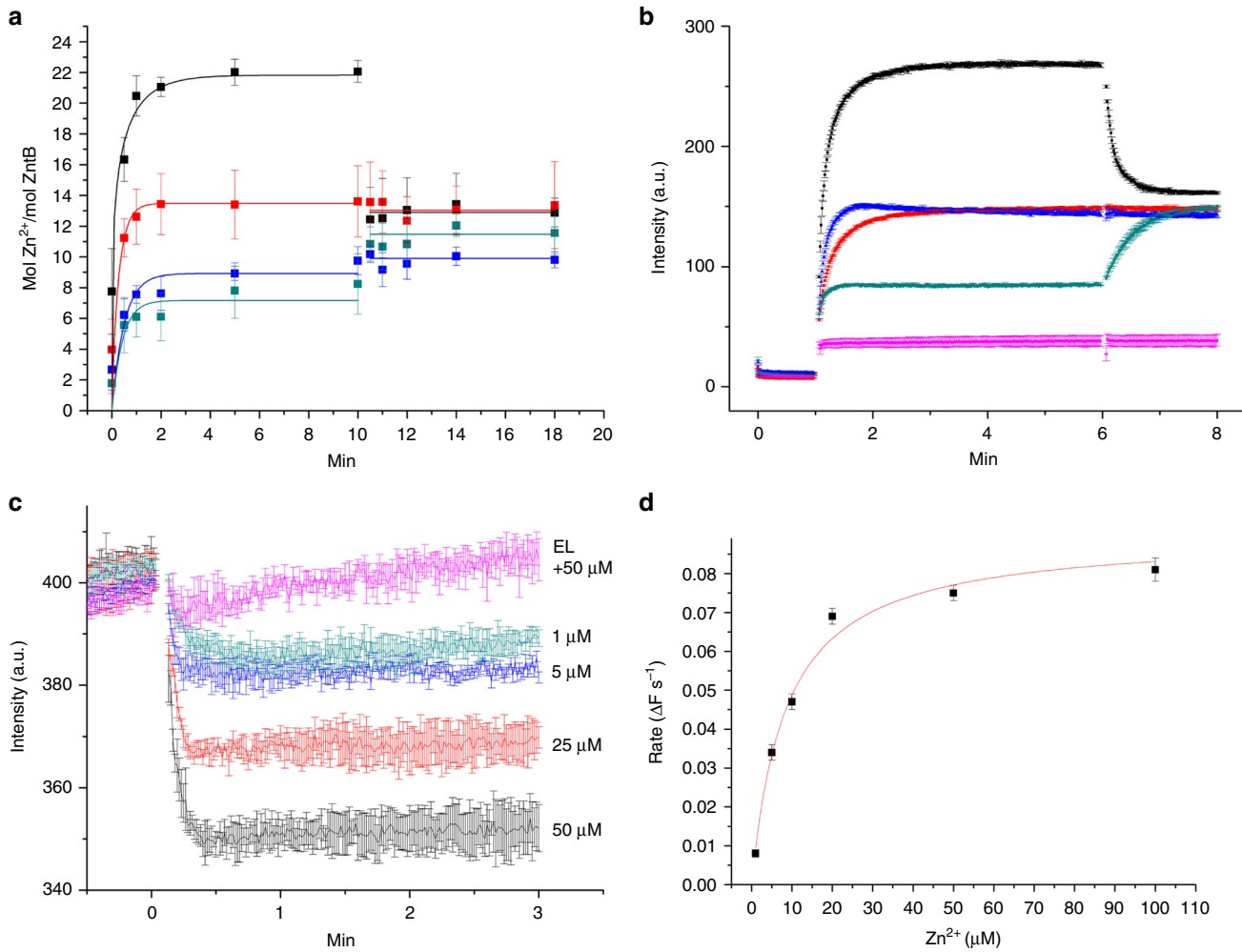

**Fig. 3** Radioactive and fluorescent transport assays (**a**) $^{65}Zn^{2+}$ uptake via ZntB reconstituted in liposomes under different conditions (colour-coded: black—inward pH flux (7.5 in/6.5 out); green—outward pH flux (6.5 in/7.5 out); red and blue—no pH flux at 6.5 and 7.5 pH, respectively). Ionophore FCCP was added at the time point of 10 min. Note the opposite effect of FCCP on direct and reverse pH gradient. Error bars represent s.e.m. from more than three technical replicates of independent batches of proteoliposomes. **b** Changes in fluorescent signal by the reporter dye FluoZin-1 during uptake of $Zn^{2+}$ (added at 1 min time point) via ZntB reconstituted in liposomes under different conditions (colour-coding as in **a**, additionally the signal from empty liposomes in magenta). FCCP was added at the time point of 6 min. Error bars represent s.e.m. from more than three technical replicates of independent batches of proteoliposomes. **c** $Zn^{2+}$-dependent transport of $H^+$ via ZntB. Quenching of the pH-dependent fluorophore ACMA at different $Zn^{2+}$ concentrations is shown by unique colours. **d** Rate of transport dependence on $Zn^{2+}$ concentration. The solid line represents the fit to the Michaelis–Menten equation with a $K_M$ of ~7.5 μM (based on FluoZin-1 experiments)

by ZntB reconstituted into the liposomes, and found that the ions that bind to the protein in ITC assay are also transported (see below).

**Zinc transport by ZntB is stimulated by pH gradient**. To characterize the mode of transport, we performed the $^{65}Zn^{2+}$ uptake assays with purified ZntB reconstituted into liposomes. Transport was measured using either equal pH on the inside and outside of the liposomes, or using pH gradients. $Zn^{2+}$ was taken up by the liposomes containing ZntB, and uptake was enhanced by a pH gradient with the lumen of the liposomes more basic than the outside. In contrast, uptake was suppressed in the presence of a reverse pH gradient. These experiments suggest that zinc transport is driven by the pH gradient. Consistently, addition of the proton ionophore FCCP at a time point when the uptake had reached a plateau led to efflux of the accumulated $^{65}Zn^{2+}$ in case of a pH gradient that was basic inside (Fig. 3a), and addition

of FCCP to liposomes with the opposite pH gradient stimulated additional uptake. FCCP did not affect the $Zn^{2+}$ accumulation in the absence of a pH gradient (Fig. 3a). These results were confirmed by performing the transport assays with the specific zinc reporting dye fluozin-1[26], encapsulated into liposomes. The fluorescence signal increased upon $Zn^{2+}$ transport into the liposomes and FCCP had a similar effect as in uptake assays with radiolabelled $Zn^{2+}$ (Fig. 3b). The observed stimulation of $Zn^{2+}$ uptake by an inward pH gradient suggests a mechanism in which protons are co-transported with $Zn^{2+}$. We directly measured $Zn^{2+}$-dependent proton transport using pH-sensitive fluorophore 9-amino-6-chloro-2-methoxyacridine (ACMA) (Fig. 3c) with proteoliposomes that had equal pH in the lumen and exterior. $Zn^{2+}$ uptake into these proteoliposomes was accompanied by generation of a pH gradient, consistent with coupled $H^+$-$Zn^{2+}$ transport mechanism. A $Na^+$ gradient instead of a proton gradient did not stimulate transport (Supplementary Fig. 7). These experiments show that $Zn^{2+}$ transport is stimulated

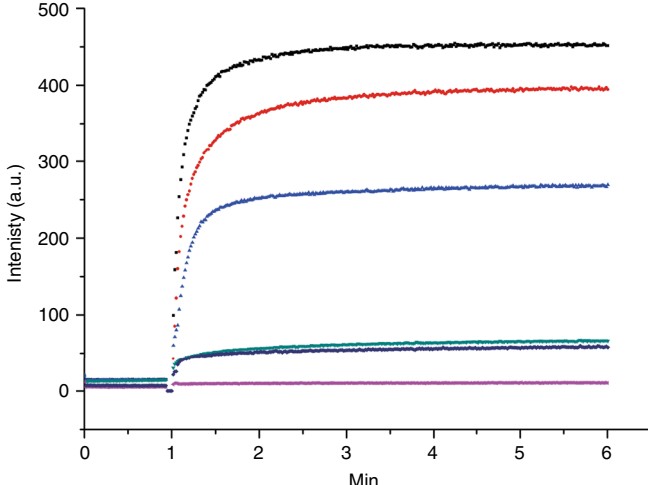

**Fig. 4** Transport of different cations by ZntB. Transport of different cations (added at 1 min and colour-coded: black—20 µM $Zn^{2+}$, red—20 µM $Cd^{2+}$, blue—100 µM $Ni^{2+}$, green—200 µM $Co^{2+}$, dark blue—empty liposomes with 20 µM $Zn^{2+}$, magenta—empty liposomes with 200 µM $Co^{2+}$) assayed by the fluorophore Fluozine-1 trapped inside the proteoliposomes

by a proton gradient across the membrane. Finally, the transport of $Zn^{2+}$ saturated with increasing $Zn^{2+}$ concentrations with a $K_M$ of ~7.5 µM (Fig. 3d), again indicative of a transporter mechanism.

To complement the ITC experiments on substrate binding (see above), we tested whether $Ni^{2+}$, $Co^{2+}$, $Cd^{2+}$ could be transported by ZntB reconstituted into the liposomes using the fluozin-1 dye, and observed comparable levels of transport for $Ni^{2+}$ and $Cd^{2+}$, but not for $Co^{2+}$. The failure to detect transport of $Co^{2+}$ is likely caused by lower sensitivity of the dye for this cation (Fig. 4).

## Discussion

ZntB is a distant homologue of CorA proteins, which are well-characterized magnesium (and cobalt) channels[13,16,17,27]. Based on whole-cell transport experiments on ZntB[12] and structures of soluble domains[14,15], it has been proposed that CorA superfamily contains both channels (CorA) and transporters (ZntB), the latter possibly using a different transport mechanism[12]. A recently reported cryo-EM structure of TmCorA in $Mg^{2+}$-free conditions, obtained upon EDTA treatment, revealed an unprecedented asymmetry of the pentamer[28]. It was concluded that CorA proteins might use a transport mechanism that involves a partial loss of the fivefold symmetry in the open state[28–30] to create a pore wide enough for the transport of partially hydrated magnesium. Our work on ZntB shows that the mechanism cannot be extrapolated to other members of the family, because even after an extensive treatment with EDTA ZntB maintained its symmetrical pentameric state (Supplementary Fig. 8). There are several possible explanations for the differences between CorA and ZntB. First, the structural differences are genuine, and related to mechanistic differences between CorA and ZntB. Second, ZntB can also form a collapsed state, but under different conditions than CorA (perhaps in the membrane). Third, the observed symmetry-collapsed state of CorA is an artefact, possibly induced by the $Mg^{2+}$-free conditions that were used to obtain the structure. Such conditions are probably never encountered by the protein under physiological conditions, as the intracellular concentration of free $Mg^{2+}$ is estimated to be around 0.5–1mM[31]. Also, other intracellular divalent and monovalent cations were shown to bind to TmCorA[30]. Therefore, it is possible that the observed symmetry-collapsed structure is not part of the

mechanism of channel opening. In contrast to stringent removal of $Mg^{2+}$ from CorA channels, the depletion of $Zn^{2+}$ from ZntB is likely to be physiologically relevant because intracellular concentrations of free $Zn^{2+}$ are extremely low (i.e., in the pM–fM range[32,33]). Therefore the symmetrical structure of ZntB may better represent the *apo* state than the asymmetrical structure of CorA.

To understand the mechanism of transport used by ZntB, the structures of the different states are essential. A comparison of our full-length structure with the structure of the soluble domain of StZntB provides a first indication of the movements that may occur within the symmetrical scaffold to provide a pathway for the transported zinc (Fig. 5). Whereas our structure was obtained in the absence of $Zn^{2+}$, StZntB was crystallized in the presence of $Zn^{2+}$. Calculation of the surface potentials revealed dramatic differences between full-length EcZntB and soluble StZntB (Fig. 5a, b). The cytoplasmic domain of full length EcZntB has a strong positive electrostatic surface potential (resembling VpZntB (Fig. 5c)), whereas the potential in the isolated domain of StZntB is negative[15]. Threading of the EcZntB sequence in the StZntB produced a similar result of more negative surface potential (Fig. 5d) and the reverse threading (StZntB sequence in EcZntB model) produced a positive surface potential (Fig. 5e). Furthermore, the shape of the internal pore between two forms is different (Fig. 5) possibly resembling two conformational states. The charge inversion of the pore surface (Fig. 5f) between the two symmetrical states might be caused by helical rotation of TM1, which bears a patch of highly conserved basic and acidic residues on the adjacent faces of the helix.

It has been debated whether ZntB is primarily used for import or export. Whole-cell assays have indicated that ZntB is a $Zn^{2+}$ and $Cd^{2+}$ exporter[12]. This conclusion was based on experiments using a knockout strain, from which it was assumed that all zinc transporters had been deleted. However, recent studies revealed that additional transporters, such as PitA[6], HoxN, ActP[34] and the STM0353 gene product (homologous to CadA[35]), might contribute significantly to zinc transport. Nevertheless, these results showed that ZntB at least affected the transport of $Zn^{2+}$ and $Cd^{2+}$. Interestingly, the analysis of regulation of ZntB expression in *C. metallidurans* revealed that it was downregulated in the presence of high concentrations of $Zn^{2+}$, $Cd^{2+}$ and $Cu^{2+}$[6], which suggests that it is an importer, rather than an exporter. Additionally the expression of homologous ZntB from *Agrobacterium tumefaciens* was not induced by treatments with $Zn^{2+}$ in a range from 100 to 750 µM[36]. Our experiments show that ZntB most likely mediates $Zn^{2+}/H^+$ co-transport, and thus indicate that ZntB is an importer for zinc. This indeed confirms that the same fold within CorA superfamily can be used either as a channel (CorA) or a transporter (ZntB).

In conclusion, we have resolved the full-length structure of a ZntB transporter, a member of the CorA MIT family. We have performed ligand binding and ligand-transport assays that unambiguously show that ZntB is involved in zinc transport. By combining all available data from us and other groups, we conclude that ZntB is a zinc importer that is driven by a proton gradient. Its transport mechanism appears distinct from that of CorA $Mg^{2+}$ channels. Unlike CorA, ZntB does not collapse into a highly asymmetrical state upon depletion of divalent cations. The elucidation of different conformational states of ZntB will be essential to describe its transport mechanism in greater detail.

## Methods

**Protein expression and membrane vesicle preparation**. Expression of ZntB was performed in a 5-l flask containing 2 l of LB medium (10 g l$^{-1}$ Bacto trypton, 5 g l$^{-1}$ Bacto yeast extract, 10 g l$^{-1}$ NaCl), supplemented with 50 µg ml$^{-1}$ kanamycin and 34 µg ml$^{-1}$ chloramphenicol. The *E. coli* BL-21(DE3) cells with pNIC_BSA4_ZntB

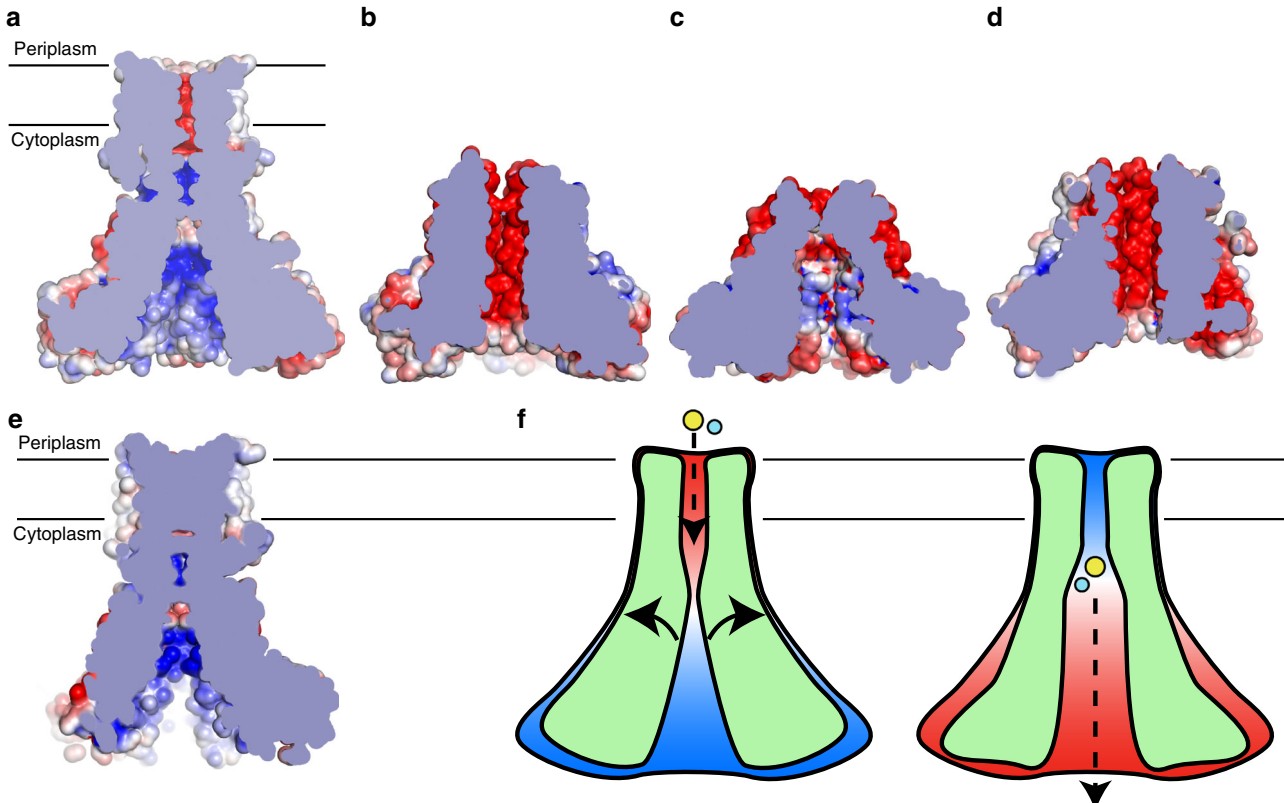

**Fig. 5** Possible mechanism of Zn$^{2+}$ transport by ZntB. **a** Calculated electrostatic potential (±5 kT e$^{-1}$) of ZntB (cross-section of the pore is shown) (**b**) same for Zn$^{2+}$-bound soluble domain of StZntB and (**c**) Zn$^{2+}$-free soluble domain of VpZntB. **d** Phyre2-based model of putative Zn$^{2+}$-bound EcZntB using StZntB as a template (pdb id 3NWI). **e** Phyre2-based model of full-length *apo* StZntB using EcZntB as a template. **f** Putative mechanism of Zn$^{2+}$ transport via ZntB. ZntB cross-section is shown schematically; Zn$^{2+}$ and H$^+$ are shown as yellow and cyan spheres, respectively; arrows indicate possible movements of trans membrane helix 1, which are possibly caused by Zn$^{2+}$ and/or H$^+$ binding, eventually leading to the change in electrostatic potential (from positive (blue) to negative (red)) within the pore that stimulates ion advancement through it

(pNIC28-Bsa4 was a gift from Opher Gileadi (Addgene plasmid #26103)[37]) were grown at 37 °C, 200 r.p.m. to an OD$_{600}$ of 0.8, with an induction by addition of 0.1 mM IPTG. After 3 h of expression, the cells were collected by centrifugation (15 min, 7446×g, 4 °C), washed in buffer A (50 mM Tris/HCl, pH 8.0) and resuspended in the buffer B (50 mM Tris/HCl, pH 8.0, 150 mM NaCl, 10 mM imidazole, 10% glycerol). Straight away, membrane vesicles were prepared or alternatively the resuspended cells were flash-frozen in liquid nitrogen and stored at −80 °C.

MgSO$_4$ of 1 mM and 50–100 μg ml$^{-1}$ DNase were added to the cells before membrane vesicle preparation. Next, the cells were disrupted and lysed by high-pressure (Constant Cell Disruption System Ltd., UK, two passages at 25 kPsi, 5 °C) cell debris was removed by low-speed centrifugation (30 min, 12,074×g, 4 °C), and membrane vesicles were collected by ultracentrifugation (120 min, 193,727×g, 4 °C). After that, the collected membrane vesicles were resuspended in buffer C (50 mM Tris/HCl, pH 8.0, 150 mM NaCl, 15% glycerol) to a final volume of 5 ml per 1 liter of cell culture. Subsequently, aliquoted membrane vesicles were flash-frozen in liquid nitrogen and stored at − 80 °C. Bradford Protein Assay (Bio-Rad) was used to determine the total protein concentration in the prepared membrane vesicles.

**Protein purification.** Prepared membrane vesicles were rapidly thawed and immediately solubilized in buffer D (50 mM Tris/HCl, pH 8.0, 150 mM NaCl, 10 mM imidazole, 10% glycerol, 1% (w/v) *n*-dodecyl-β-D-maltopyranoside (DDM, Anatrace)) at 4 °C for 1 h, while gently rocking. To remove unsolubilized material the centrifugation step (30 min, 442,907×g, 4 °C) was applied. After that, the supernatant was incubated for 1 h with Ni$^{2+}$-sepharose resin (column volume of 0.5 ml) at 4 °C, pre-equilibrated with 20 CV of buffer E (50 mM Tris/HCl, pH 8.0, 150 mM NaCl, 15 mM imidazole, 0.03% DDM). Next, the flow through was collected after the suspension had been poured into a 10-ml disposable column (Bio-Rad). The column material was washed with 10 ml of buffer E. ZntB was eluted in three fractions of buffer F (50 mM Tris/HCl, pH 8.0, 250 mM NaCl, 500 mM imidazole, 0.03% (w/v) DDM) of 200, 750 and 500 μl, respectively. The second elution fraction was treated with 2 mM of EDTA to remove co-eluted Ni$^{2+}$ ions and any residual zinc. Later on, the second elution fraction was subjected to size-exclusion chromatography using a Superdex 200 10/300 gel filtration column

(GE-Healthcare), pre-equilibrated with buffer G (50 mM Tris/HCl, pH 8.0, 250 mM NaCl, 0.03% (w/v) DDM). Fractions containing purified were combined and used directly for proteoliposome reconstitution, or concentrated by the use of a Vivaspin 500 concentrating device with a molecular weight cutoff of 100 kDa (Sartorius stedim) to a final concentration of 3–6 mg ml$^{-1}$ with or without additional 1 mM EDTA added when prepared for Cryo-EM.

**Reconstitution into proteoliposomes.** Reconstitution in proteoliposomes was performed as follows (please see ref. [38] for details): polar lipids of *E. coli* and egg phosphatidylcholine (in 3:1 (w/w) ratio) were dissolved in chloroform, then dried in a rotary evaporator and subsequently resuspended in buffer containing 50 mM KPi, pH 7.5 to the concentration of 20 mg ml$^{-1}$. After three freeze-thaw cycles, large unilamellar vesicles (LUVs) were obtained and stored in liquid nitrogen. To prepare proteoliposomes, LUVs were extruded through a 400-nm-diameter poly-carbonate filter (Avestin, 11 passages). Obtained liposomes were diluted to 4 mg ml$^{-1}$ in buffer H (50 mM HEPES, pH 7.5) or buffer I (50 mM HEPES, pH 6.5) and subsequently destabilized beyond $R_{sat}$ with Triton X-100. Purified ZntB was added to the liposomes at a weight ratio of 1:250 (protein/lipid), followed by detergent removal using Bio-beads (50 mg ml$^{-1}$, four times after 0.5 h, 1 h, 2 h and overnight incubation). Afterwards, proteoliposomes were collected by centrifugation (25 min, 285,775×g, 4 °C) and resuspended in buffer H or buffer I to a lipid concentration of 10 mg ml$^{-1}$. Finally, after three freeze-thaw cycles, obtained proteoliposomes were stored in liquid nitrogen until subsequent experiments.

**Radiolabelled $^{65}$Zn$^{2+}$ transport assay.** To use in the transport assay, proteoliposomes with desired pH were thawed and extruded through a 400-nm pore size polycarbonate filter (Avestin, nine passages). Subsequently, two active units of ProTev Plus (Promega) were added to the protein sample and incubated overnight. The proteoliposomes were diluted 10 times to a final volume of 2 ml in the same buffer. Following centrifugation step (25 min, 285,775×g, 4 °C), the proteoliposomes were resuspended in buffer H or I to a final concentration of 0.5 μg ul$^{-1}$ ZntB. For each time point in the transport assays, a reaction volume of 200 μl of buffer (with desired pH) with 22 μM of $^{65}$ZnCl$_2$ added, was incubated at 30 °C while being stirred. Transport was initiated by adding 1 μg of ZntB, previously reconstituted in proteoliposomes. Stop buffer of 2 ml (ice-cold outside buffer)

| Table 1 Data collection and refinement statistics | |
| --- | --- |
| **Data collection** | |
| Microscope | Titan KRIOS with K2-detector |
| Voltage | 300 kV |
| Pixel size (Å) | 1.43 |
| Micrographs collected (#) | 2655 |
| **Refinement** | |
| Particles (#) | 333,490 |
| Resolution (Å; at FSC = 0.143) | 4.2 |
| CC (model to map fit) | 0.81 (0.83) |
| **RMS deviations** | |
| Bonds (Å) | 0.007 |
| Angles (°) | 1.152 |
| Chirality (°) | 0.065 |
| Planarity (°) | 0.008 |
| **Validation** | |
| Clash score | 12 |
| Favoured rotamers (%) | 98.77 |
| Ramachandran favoured (%) | 91.69 |
| Ramachandran allowed (%) | 8.31 |
| Ramachandran outliers (%) | 0 |

was added at the indicated time point, and the reaction was rapidly filtered over a BA-85 nitrocellulose filter. After the filter was washed with another 2 ml of stop buffer, the levels of radioactivity were determined using a Packard Cobra II 5010 Gamma counter.

**Fluorescent transport assays**. Zinc transport was measured with the $Zn^{2+}$-sensitive fluorophore FluoZin-1 (ThermoFisher, USA). To avoid bleaching of the fluorophore, the sample was shielded from the direct light as much as possible. FluoZin-1 (stock concentration 3 mM in $H_2O$) was added to a final concentration of 5 μM to the proteoliposomes with desired pH. FluoZin-1 encapsulation was performed by three freeze–thaw cycles and subsequent extrusion through 0.4-μm polycarbonate filters. Extravesicular dye was removed from ~500 μl of liposome suspension by size exclusion chromatography on a 2 ml Sephadex G-75 column equilibrated with buffer H or I. Proteoliposomes were collected by ultracentrifugation (25 min, 285,775×g, 4 °C), and the supernatant was removed. Proteoliposomes were resuspended with 10 μl buffer H or I per 2.5 mg of proteoliposomes (protein to lipid ratio 1:250). Transport assays with or without proton gradient were initiated by the addition of 10 mM stock solution of zinc acetate to the desired final concentration. For each measurement, 0.3 mg of proteoliposomes (protein to lipid ratio 1:250) was diluted in 1 ml of desired buffer. A fluorescence time course was measured in a 1 ml cuvette with a stirrer using an excitation wavelength of 490 nm and an emission wavelength of 525 nm. Experiments with empty liposomes were performed in parallel as controls. Initial transport rates ($\Delta F\, s^{-1}$) were calculated by performing a linear regression on the transport data between 1 and 10 s after addition of zinc acetate. The resulting data was fitted to a Michaelis–Menten equation. All measurements were at least triplicated.

For $H^+$ transport assays, the lumenal buffer of the proteoliposomes was exchanged for buffer J (5 mM HEPES pH 6.7) by resuspension of the liposomes in this buffer followed by three freeze-thaw cycles and extrusion through 0.4-μm polycarbonate filters. Proteoliposomes were collected by ultracentrifugation (25 min, 285,775×g, 4 °C), and the supernatant was removed. Proteoliposomes were resuspended with 10 μl buffer J per 2.5 mg of proteoliposomes (protein to lipid ratio 1:250). For each measurement, 0.3 mg of proteoliposomes was diluted in 1 ml of buffer K (5 mM HEPES, pH 6.7, 150 nM ACMA). A fluorescence time course was measured in a 1 ml cuvette with a stirrer using an excitation wavelength of 419 nm and an emission wavelength of 483 nm; zinc was added after 3 min of equilibration time. Experiments with empty liposomes were performed in parallel as controls. All measurements were triplicated.

**Isothermal titration calorimetry**. ITC200 instrument (MicroCal) was used to perform all ITC experiments. The thermally equilibrated ITC cell was filled with 280 μl of ZntB in buffer G (10–15 μM) and studied substrates (in the same buffer) were titrated into the cell. Temperature was fixed at 25 °C. Analysis of data was performed using the origin-based software provided by MicroCal.

**Single particle cryo-electron microscopy**. The purified ZntB sample was adjusted to a final concentration of ~10 mg ml$^{-1}$. Aliquots of 3 μl were applied to a freshly glow-discharged holey carbon grids (Quantifoil Au R1.2/1.3, 300 mesh), excess liquid was blotted for 4–5 s using a FEI Vitrobot Mark IV and the sample was

plunge frozen in liquid ethane at a temperature of approximately 100 K. TEM grids were transferred into a Titan Krios 300 keV microscope (FEI, Netherlands), equipped with a K2 direct-electron detector. Zero-loss images were recorded semi-automatically, using the UCSF Image4 script[39]. The GIF-quantum energy filter was adjusted to a slit width of 20 eV. Images were collected at a nominal magnification of ×81,000 (yielding a pixel size of 1.43 Å) and a defocus range of −1.5 to −3.0 μm. A total of 2655 movie images were collected with 24 frames dose-fractionated over 18 s, in super-resolution counting mode.

Beam-induced motion in the raw movie frames was corrected for using whole-frame motion correction with MOTIONCORR 1.0[40], followed by contrast transfer function (CTF) estimation using gctf[41]. All subsequent data processing steps were performed using the RELION 2.0 software suite[42]. References for template-based particle picking were obtained from two-dimensional (2D) classes obtained from manually picked particles from a subset of micrographs. The initial run of template-based algorithm picked 1 million particles from all 2655 images. To reduce the number of false-positive particle picks from the initial template-based particle picking, several rounds of 2D classification were applied to the full extracted data set, resulting in a subset of 333,490 particle projections. The resulting particles were submitted to three-dimensional auto-refinement, particle-based motion correction and damage-based weighting of individual frames[43]. An additional round of 2D classification was performed on these 'polished' particles to further discard false-positive or low-quality particles. The obtained map was used for manual model building in Coot[44] using the previously published CorA structure (pdb id: 4I0U) as a reference model. Refinement was performed in Phenix[45] with the final validation check in Molprobity[46]. The '$Zn^{2+}$-bound' model of EcZntB and '$Zn^{2+}$-free' model of StZntB were obtained by threading EcZntB sequence into StZntB model using the Phyre2 server[47]. Electrostatic potentials were calculated with APBS[48] after the initial preparation of files at PDB2PQR server[49]. Images were prepared with the open source version of PyMol (https://sourceforge.net/projects/pymol/).

**Data availability**. Atomic coordinates and the corresponding electron microscopy density map are deposited in the Protein Data Bank and the Electron Microscopy Data Bank under accession number 5N9Y and EMD-3605, respectively. Other data are available from the corresponding authors upon reasonable request.

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

## Acknowledgements

We are grateful to M. Guskova for help with figures preparation. This research was supported by HFSP fellowship (LTF000087/2015-L) to C.G. and NWO Vidi grant 723.014.002 to A.G.

## Author contributions

A.G. conceived the project. Expression, purification, ITC, radioactive zinc uptakes and transport assays were performed by A.S. Cryo-EM was carried out by C.G. and S.H.W.S. Model building and refinement was done by A.G. A.S., C.G., D.J.S., S.H.W.S. and A.G. analysed the data. A.G. and D.J.S. wrote the manuscript with input from all other authors.

## Additional information

**Competing interests:** The authors declare no competing financial interests.

