## [Peer Review File · Nature Communications]

Reviewers' comments:

Reviewer #1 (Remarks to the Author):

This work aims to elucidate the molecular basis for ZntB-mediated zinc transport. Overall I think this work is of potential interest and the data are clearly presented. I have summarized my suggestions about this manuscript as follows.

1) The authors have established several assays to study the zinc transport by ZntB. It seems that it will be feasible to study the kinetics of zinc transport using their assays. However, the authors have not utilized such kinetic studies to determine whether ZntB is a channel or a transporter. In order to discuss the transport mechanism in depth, I believe the authors need to address this issue about channel vs. transporter experimentally.

2) The authors presented data on ITC, which presumably will reveal the stoichiometry between Zn^{2+} and ZntB. Such important analysis or discussion is missing in the manuscript. This stoichiometry may also shed light on the issue about channel vs. transporter.

3) The authors suggested that ZntB selects cations based on their size. For lay readers, it is difficult to follow the authors since the ionic radii were not discussed in the manuscript. Besides size, how about cation coordination geometry? It seems that size is unlikely to be the only factor that contributes to cation selectivity in ZntB. Such discussion should be added to or expanded in the manuscript to make the story more complete. Can the authors see any density for bound cations in ZntB?

4) I think it would enhance the manuscript if the authors can discuss how ZntB utilizes the proton gradient transport zinc and where the protonation site may be located in ZntB, so that a more coherent mechanism can emerge from this work.

Reviewer #2 (Remarks to the Author):

The manuscript of Gati et al. describes the structural and functional characterization of ZntB, a membrane transport protein involved in zinc homeostasis in prokaryotes. ZntB is a member of

the Metal Ion Transporter family of pentameric funnel-shaped channels from which the magnesium transporter CorA is the best characterized member concerning both structure and function. Importantly, the zinc transporters of the MIT family are part of a very different branch as the magnesium transporters with a corresponding low sequence identity. The major findings of this manuscript are the new structure of a complete MIT protein with different substrate specificity and the suggestion of a secondary proton-coupled transport mechanism instead of channel/facilitated diffusion type of transport.

The manuscript presents the first structure of a full-length protein from the zinc transporter branch of the MIT family obtained using single-particle cryo-electron microscopy and at an overall resolution of 4.2 Angstrom. Thus far, only high resolution structures of the cytoplasmic domains of zinc transporters in different conformations were known (Tan et al., 2009: VpZntB with tapered funnel; Wan et al., 2011: StZntB with cylindrical funnel). The description of a complete zinc transporter is an important and novel finding, but a relevant question to be asked is to what extent this novel structure leads to a significant advance in our understanding of these proteins. The different conformation of the EcZntB cytoplasmic domain with respect to that of StZntB is in that sense not a major advance, as the occurrence of this potential conformational change was already clear from comparing StZntB with VpZntB. The structure of a full-length transporter would allow addressing how this potential conformational change in the cytoplasmic domain affects the membrane domain, but this would require an additional structure of the complete transporter, possibly to be obtained in the presence of zinc (so under similar conditions as used to crystallize the cytoplasmic domain of StZntB). Regrettably, this structure is not available. Alternatively, the current structure could be used to address the interesting ion selectivity, but despite the availability of functional assays and a nice structure the authors do not look into this point.

In addition, the authors demonstrate that zinc ion transport by ZntB is stimulated by an inward-directed pH gradient, suggesting that ZntB is a coupled secondary transporter and not a channel/facilitator. For reasons unclear to me, this finding is presented as a comparably small result, but, provided it is true, I feel it should receive much more appreciation as existing literature suggests these proteins function exclusively as channels/facilitators. It should also require additional validation. I encourage the authors to discuss this observation to a much greater extent. A few points: Concerning the (heroic) ^{65}Zn transport assay: all curves seem to converge to the origin, but the basis for this (apart from a theoretical) is not clear. Did the authors take 0 min timepoints and did they use these to subtract the background? The first timepoints depicted seem to be 0 min values, but these differ significantly. If these were to be used for background subtraction the observed differences before addition of the protonophore seem to be nullified. Please comment. In addition, an empty liposome background sample is missing. Apart from this, the 2-3X accumulation in the presence of an inward pH gradient is low, but provided the background subtraction was done correctly, significant. The complementary fluorescence uptake is very convincing. Additional investigations should include an experiment where zinc-driven proton uptake is studied using pH indicators such as ACMA or pyranine. It should also

demonstrate whether an inward sodium gradient could drive zinc transport. It should also discuss, and ideally demonstrate, which residue(s) is/are involved in proton coupling. Concluding, I greatly appreciate the novel structure, the functional characterization of the protein, and the options these data have to further the mechanistic understanding of MIT transporters. In its current form, however, the manuscript does not appropriately address these questions that would broaden its scope beyond a specialist community. At the same time, the manuscript has the potential to become an important paper that might influence our thinking on these proteins. I would like to encourage the authors to address the above-mentioned points in a large revision.

Additional remarks

Page5 Does threading of the StZntB on EcZntB also lead to a reverse of negative surface charge to positive?

Page6 Discussion on directionality of transport, “with caution”. This is too strongly formulated. In the Worlock paper an increased efflux is observed compared to the knock-out. Even if this knock-out contains additional zinc transporters this is still a good indication that ZntB can facilitate export (as well). If ZntB is a secondary transporter, for which evidence has been presented in the manuscript, transport should in principle be able to proceed in both directions with the directionality at any moment in time depending on the sum of the relevant gradients. In vivo, the pH and electrical gradients would support import indeed. Please include a similar line of argumentation in this section to support the statement in the final sentence.

Minor remarks

Fig3a the symbols should be chosen so that it is intuitively clearer that the control for in7.5/out6.5 is in7.5/out7.5, e.g. by using identical symbols with the control not-filled. The current color/symbol scheme is prone to misinterpretation.

Fig4d legend “phyre2-based model”, indicate what template was used here

Fig4e function of arrows in the left cartoon are unclear. Based on the available structures the right cartoon should show a wider cylinder in the cytoplasmic side. The legend describing the colors is not clear (“red and blue indicate possible change..”)

Page5 the section addressing the asymmetry should be worded more carefully. The fact that no asymmetric state was observed cannot exclude that such a state is not occurring, e.g. in the context of a lipid membrane. Also correct the relevant statement in the summary section of the discussion.

Page8 “2AU ProTEV plus” please clarify what was done.

Page9 the zinc concentration used for the fluorescent transport assay is not indicated. To what extent does the experimental outcome depend on the use of zinc-acetate instead of zinc-chloride (given the membrane-permeability of acetic acid)? Please comment

The methods section is full of small errors and seems to be written in a hasty manner (storage of vesicles at 80 degrees should be -80. Elution volumes from the IMAC column of 200-750 milliliter should be microliter. Furthermore, on several positions super-/subscripts, italics, and

spaces are missing.

Reviewer #3 (Remarks to the Author):

In this manuscript by Guskov and colleagues, authors solved a structure of the zinc transporter ZntB by cryo-EM, proposed that ZntB is a Zn²⁺ importer, and suggested that mechanism of transport distinct from CorA (Mg²⁺ channel). This manuscript is well written and all the techniques that have been utilized in this study are well established. This is a first structural and mechanistical understanding of ZntB importer, thus the significance of the manuscript is apparent. However, for this manuscript to be considered for publication, the authors would need to work on the manuscript data representation and address these concerns.

- 1) One of the major conclusions of the paper is that ZntB does not use the same mechanism of transport as CorA. This is very intriguing conclusion to make, so authors should not hide their cryo-EM structure of ZntB with 1mM EDTA in Extended Figure 7, but show this structure and compare it to the ZntB without 1mM EDTA in the main figure together with data from published CorA studies. This would be much easier to understand their major conclusions.
- 2) Authors should also include cryo-EM data analysis information such as representative micrograph, representative particle images, 2D classes, a summary of the image processing procedure and etc. for ZntB with and without 1mM EDTA.
- 3) I am not sure if all the information provided in Figure 2 and Figure 3 should be in the main text. Can authors combine some information for the main figure and keep the rest in Extended Figures?
- 4) Overall, main text figures and some extended figures should be re-done/combined/improved to better understand the paper and paper conclusions.

Reviewer #1 (Remarks to the Author):

This work aims to elucidate the molecular basis for ZntB-mediated zinc transport. Overall I think this work is of potential interest and the data are clearly presented. I have summarized my suggestions about this manuscript as follows.

1) The authors have established several assays to study the zinc transport by ZntB. It seems that it will be feasible to study the kinetics of zinc transport using their assays. However, the authors have not utilized such kinetic studies to determine whether ZntB is a channel or a transporter. In order to discuss the transport mechanism in depth, I believe the authors need to address this issue about channel vs. transporter experimentally.

We have now included new experimental data that indicate a transport mechanism. First we have shown that transport rates saturate at increasing Zn^{2+} concentration (Figure 3d), consistent with transport rather than channel activity. Second, we have measured proton flux associated with Zn^{2+} transport using the fluorescent dye ACMA (Figure 3c).

2) The authors presented data on ITC, which presumably will reveal the stoichiometry between Zn^{2+} and ZntB. Such important analysis or discussion is missing in the manuscript. This stoichiometry may also shed light on the issue about channel vs. transporter.

We have now included this information. ITC experiments showed 1:1 stoichiometry, but we are not sure whether this information helps to resolve the transporter vs channel question. We believe other data (mentioned in the point (1) give a better indication that it is a transporter.

3) The authors suggested that ZntB selects cations based on their size. For lay readers, it is difficult to follow the authors since the ionic radii were not discussed in the manuscript. Besides size, how about cation coordination geometry? It seems that size is unlikely to be the only factor that contributes to cation selectivity in ZntB. Such discussion should be added to or expanded in the manuscript to make the story more complete. Can the authors see any density for bound cations in ZntB?

We now have extended a discussion about possible determinants of selectivity (size, hydration shell coordination, water residence time in hydration shells), however still there is no clear understanding how the selectivity is achieved. Since the protein sample was treated extensively with EDTA, we do not expect to have any zinc bound, but in any case it is impossible to unambiguously assign ions in cryo-EM structures (in contrast to X-ray crystallography, where it can be easily checked with anomalous diffraction).

4) I think it would enhance the manuscript if the authors can discuss how ZntB utilizes the proton gradient transport zinc and where the protonation site may be located in ZntB, so that a more coherent mechanism can emerge from this work.

We now include the data for the proton coupling based on ACMA fluorescence (see also response to reviewer #2).

Reviewer #2 (Remarks to the Author):

The manuscript of Gati et al. describes the structural and functional characterization of ZntB, a membrane transport protein involved in zinc homeostasis in prokaryotes. ZntB is a member of the Metal Ion Transporter family of pentameric funnel-shaped channels from which the magnesium transporter CorA is the best characterized member concerning both structure and function. Importantly, the zinc transporters of the MIT family are part of a very different branch as the magnesium transporters with a corresponding low sequence identity. The major findings of this manuscript are the new structure of a

complete MIT protein with different substrate specificity and the suggestion of a secondary proton-coupled transport mechanism instead of channel/facilitated diffusion type of transport. The manuscript presents the first structure of a full-length protein from the zinc transporter branch of the MIT family obtained using single-particle cryo-electron microscopy and at an overall resolution of 4.2 Angstrom. Thus far, only high resolution structures of the cytoplasmic domains of zinc transporters in different conformations were known (Tan et al., 2009: VpZntB with tapered funnel; Wan et al., 2011: StZntB with cylindrical funnel). The description of a complete zinc transporter is an important and novel finding, but a relevant question to be asked is to what extent this novel structure leads to a significant advance in our understanding of these proteins. The different conformation of the EcZntB cytoplasmic domain with respect to that of StZntB is in that sense not a major advance, as the occurrence of this potential conformational change was already clear from comparing StZntB with VpZntB.

The previous structures of the cytoplasmic domains of ZntB could not explain the transport mechanism, as the crucial transmembrane part of the protein was missing. Even though the full-length structure looks similar to Mg²⁺ bound CorA the (transmembrane) pore lining residues are not as predicted (e.g. there are no His residues), and the pore itself has a different geometry. Furthermore, the full-length structure revealed that the selectivity filter is well conserved among even distant members of MIT family (sequence identities between CorAs and ZntBs are below 15-20%). Finally, the full-length structure of ZntB shows that the collapse of symmetry which was observed in substrate-stripped CorA does not occur in ZntB. All these findings contribute to the news value.

The structure of a full-length transporter would allow addressing how this potential conformational change in the cytoplasmic domain affects the membrane domain, but this would require an additional structure of the complete transporter, possibly to be obtained in the presence of zinc (so under similar conditions as used to crystallize the cytoplasmic domain of StZntB). Regrettably, this structure is not available.

We completely agree with this reviewer that the zinc-bound structure would help tremendously, but it is technically very difficult, because Zn²⁺ interferes with the preparation of the EM grids. We hope to solve these technical problems in the future, but for now it is not possible to obtain high quality single particle EM data on ZntB in the presence of Zn²⁺.

Alternatively, the current structure could be used to address the interesting ion selectivity, but despite the availability of functional assays and a nice structure the authors do not look into this point.

We now include transport assays with different cations in the manuscript (Figure 4) and show that Cd²⁺, Ni²⁺ and (possibly) Co²⁺ are also transported substrates.

In addition, the authors demonstrate that zinc ion transport by ZntB is stimulated by an inward-directed pH gradient, suggesting that ZntB is a coupled secondary transporter and not a channel/facilitator. For reasons unclear to me, this finding is presented as a comparably small result, but, provided it is true, I feel it should receive much more appreciation as existing literature suggests these proteins function exclusively as channels/facilitators. It should also require additional validation. I encourage the authors to discuss this observation to a much greater extent.

We agree with this reviewer and now we emphasize this finding to a larger extent. We have included additional experiments to validate that protons are co-transported with Zn²⁺.

A few points: Concerning the (heroic) 65Zn transport assay: all curves seem to converge to the origin, but the basis for this (apart from a theoretical) is not clear. Did the authors take 0 min timepoints and did they use these to subtract the background? The first timepoints depicted seem to be 0 min values, but these differ significantly. If these were to be used for background subtraction the observed differences before addition of the protonophore seem to be nullified. Please comment.

We subtracted the empty liposome background. The first timepoints are 5 seconds and have quite big error bars.

In addition, an empty liposome background sample is missing. Apart from this, the 2-3X accumulation in the presence of an inward pH gradient is low, but provided the background subtraction was done correctly, significant.

The empty liposome background was subtracted.

The complementary fluorescence uptake is very convincing. Additional investigations should include an experiment where zinc-driven proton uptake is studied using pH indicators such as ACMA or pyranine.

We now include the assay with ACMA that shows proton coupling to zinc transport.

It should also demonstrate whether an inward sodium gradient could drive zinc transport.

A sodium gradient could not drive the transport. The result is now included in the supplementary information (Supplementary Fig. 7).

It should also discuss, and ideally demonstrate, which residue(s) is/are involved in proton coupling.

Unfortunately structures do not always provide an immediate answer to this question, and there are several examples where assignment of exact residues to which proton coupling occurs took a very long time or haven't been finished yet – LeuT transporters, BetP proteins, glutamate transporters, etc. It is on our to-do list, and at this point we would rather not speculate, but do a meticulous (separate) study in future.

Concluding, I greatly appreciate the novel structure, the functional characterization of the protein, and the options these data have to further the mechanistic understanding of MIT transporters. In its current form, however, the manuscript does not appropriately address these questions that would broaden its scope beyond a specialist community. At the same time, the manuscript has the potential to become an important paper that might influence our thinking on these proteins. I would like to encourage the authors to address the above-mentioned points in a large revision.

Additional remarks

Page5 Does threading of the StZntB on EcZntB also lead to a reverse of negative surface charge to positive?

Yes, we now include this result as an additional panel (e) in Figure 5.

Page6 Discussion on directionality of transport, “with caution”. This is too strongly formulated. In the Worlock paper an increased efflux is observed compared to the knock-out. Even if this knock-out contains additional zinc transporters this is still a good indication that ZntB can facilitate export (as well). If ZntB is a secondary transporter, for which evidence has been presented in the manuscript, transport should in principle be able to proceed in both directions with the directionality at any moment in time depending on the sum of the relevant gradients. In vivo, the pH and electrical gradients would support import indeed. Please include a similar line of argumentation in this section to support the statement in the final sentence.

We completely agree with this and we have modified the discussion accordingly.

Minor remarks

Fig3a the symbols should be chosen so that it is intuitively clearer that the control for in7.5/out6.5 is in7.5/out7.5, e.g. by using identical symbols with the control not-filled. The current color/symbol scheme is prone to misinterpretation.

We corrected the representation, but we would like to point out that these are all separate experiments with unique conditions and not really experiments+controls. To avoid confusion, we removed other symbols, but we prefer to keep the color-coding as it helps to compare these results with those of the fluorescence uptakes in panel b.

Fig4d legend “phyre2-based model”, indicate what template was used here

Done.

Fig4e function of arrows in the left cartoon are unclear. Based on the available structures the right cartoon should show a wider cylinder in the cytoplasmic side. The legend describing the colors is not clear (“red and blue indicate possible change..”)

Fixed and explained.

Page5 the section addressing the asymmetry should be worded more carefully. The fact that no asymmetric state was observed cannot exclude that such a state is not occurring, e.g. in the context of a lipid membrane.

We have rephrased this part. We believe that the fact we do not observe such an asymmetry is an important piece of information necessary for better understanding of transport mechanisms of these proteins (also highlighted by the reviewer #3), especially highlighting differences between CorA and ZntB.

Also correct the relevant statement in the summary section of the discussion.
Page8 "2AU ProTEV plus" please clarify what was done.

Corrected.

Page9 the zinc concentration used for the fluorescent transport assay is not indicated. To what extent does the experimental outcome depend on the use of zinc-acetate instead of zinc-chloride (given the membrane-permeability of acetic acid)? Please comment

The concentration is included now. We did not see any significant differences between different zinc salts.

The methods section is full of small errors and seems to be written in a hasty manner (storage of vesicles at 80 degrees should be -80. Elution volumes from the IMAC column of 200-750 milliliter should be microliter. Furthermore, on several positions super-/subscripts, italics, and spaces are missing.

We thank this reviewer for a very thorough check, and we ask for an excuse for these typos. We believe that now we have corrected all of them.

Reviewer #3 (Remarks to the Author):

In this manuscript by Guskov and colleagues, authors solved a structure of the zinc transporter ZntB by cryo-EM, proposed that ZntB is a Zn²⁺ importer, and suggested that mechanism of transport distinct from CorA (Mg²⁺ channel). This manuscript is well written and all the techniques that have been utilized in this study are well established. This is a first structural and mechanistical understanding of ZntB importer, thus the significance of the manuscript is apparent. However, for this manuscript to be considered for publication, the authors would need to work on the manuscript data representation and address these concerns.

1) One of the major conclusions of the paper is that ZntB does not use the same mechanism of transport as CorA. This is very intriguing conclusion to make, so authors should not hide their cryo-EM structure of ZntB with 1mM EDTA in Extended Figure 7, but show this structure and compare it to the ZntB without 1mM EDTA in the main figure together with data from published CorA studies. This would be much easier to understand their major conclusions.

We apologize for the confusion. Both structures were obtained with EDTA (described in methods), the one shown in supplementary underwent an additional treatment (in addition to 2mM EDTA used during the purification extra 1mM EDTA was added before vitrification), to make sure that all Zn²⁺ had been captured. But in fact both structures are very similar. We now explicitly state in the main text that the protein sample was treated with EDTA.

2) Authors should also include cryo-EM data analysis information such as representative micrograph, representative particle images, 2D classes, a summary of the image processing procedure and etc. for ZntB with and without 1mM EDTA.

Since the structures are the same (see point 1), we believe it is redundant to do so.

3) I am not sure if all the information provided in Figure 2 and Figure 3 should be in the main text. Can authors combine some information for the main figure and keep the rest in Extended Figures?

4) Overall, main text figures and some extended figures should be re-done/combined/improved to better understand the paper and paper conclusions.

We have redone some of the figures to include additional experiments requested by other reviewers: now Figure 3 additionally includes data from proton transport assay measured with ACMA dye and calculations of Km value; Figure 4 now shows the fluorescence transport assays of different cations, in Figure 5, there is an additional panel showing the model of *apo* full-length structure of StZntB and Supplementary Fig. 7 shows inability of sodium gradient to drive the transport. Since transport assays

are equally important as the structure, we prefer keeping Figure 2 and 3 in the main text. In addition reviewers 1 and 3 found this data very important, therefore we believe their presence in the main text is justifiable.

Reviewers' Comments:

Reviewer #1 (Remarks to the Author):

Overall I think the manuscript has been improved. That said, a number of issues deserve to be further addressed and are listed below.

1) It is my understanding that in proteins zinc ion generally prefers tetrahedral, rather than octahedral coordination arrangement (line 85). In the zinc transporter YjiP, for instance, the coordination for the observed zinc ions is tetrahedral (ref. 11). Cadmium ion can also have tetrahedral coordination and often competes with the zinc ion. Therefore, I think the statement in line 85 is not entirely correct.

2) In light of the consideration specified in 1), how do the authors explain the observation that ZntB binds/transport cobalt/nickel, but not magnesium ion? Based on the explanations that the authors provide (line 83), ZntB should also transport the magnesium ion but actually it does not. How about the cadmium ion, can ZntB bind and transport cadmium ion, which is commonly used to study the selectivity of zinc transporters?

3) The authors avoid discussing the specific protonation sites in ZntB. However, I think that it would be important to at least describe in general how ZntB couples the uptake of proton and zinc ion, since it seems rather difficult to comprehend how ZntB functions if the authors leave out proton in the transport mechanism (Fig. 5).

4) The authors refer to both decylmaltoside (line 54) and dodecylmaltoside (line 223) as DDM, which is confusing.

Reviewer #2 (Remarks to the Author):

My questions have been appropriately addressed. The additional experiments further strengthen the claim that the protein is a proton-coupled secondary transporter.

Minor remark:

fig3d From what measurements were the initial rates for the Michaelis-Menten plot taken?
Radioactive, Zn-fluor, ACMA?

Reviewer #3 (Remarks to the Author):

Authors addressed my comments.

Paper is suitable for publication.

In this file we provide point to point responses to the reviewers (highlighted in yellow).

Reviewers' comments:

Reviewer #1 (Remarks to the Author):

Overall I think the manuscript has been improved. That said, a number of issues deserve to be further addressed and are listed below.

1) It is my understanding that in proteins zinc ion generally prefers tetrahedral, rather than octahedral coordination arrangement (line 85). In the zinc transporter YjiP, for instance, the coordination for the observed zinc ions is tetrahedral (ref. 11). Cadmium ion can also have tetrahedral coordination and often competes with the zinc ion. Therefore, I think the statement in line 85 is not entirely correct.

We discuss here the coordination of given cations in their solvated state (with bound water molecules) and not yet bound to a protein, and in water solution these cations prefer octahedral coordination. We added this information on page 5.

2) In light of the consideration specified in 1), how do the authors explain the observation that ZntB binds/transport cobalt/nickel, but not magnesium ion? Based on the explanations that the authors provide (line 83), ZntB should also transport the magnesium ion but actually it does not. How about the cadmium ion, can ZntB bind and transport cadmium ion, which is commonly used to study the selectivity of zinc transporters?

We discuss this in lines 92-96, and unfortunately there is no irrefragable answer yet why CorAs cannot transport Zn^{2+} and ZntB cannot transport Mg^{2+} . For Cd^{2+} we show both ITC and fluorescent transport data (Figures 2 and 4 respectively).

3) The authors avoid discussing the specific protonation sites in ZntB. However, I think that it would be important to at least describe in general how ZntB couples the uptake of proton and zinc ion, since it seems rather difficult to comprehend how ZntB functions if the authors leave out proton in the transport mechanism (Fig. 5).

To avoid the confusion we added H^+ to the Figure 5

4) The authors refer to both decylmaltoside (line 54) and dodecylmaltoside (line 223) as DDM, which is confusing.

We corrected the typo

Reviewer #2 (Remarks to the Author):

My questions have been appropriately addressed. The additional experiments further strengthen the claim that the protein is a proton-coupled secondary transporter.

Minor remark:

fig3d From what measurements were the initial rates for the Michaelis-Menten plot taken? Radioactive, Zn-fluor, ACMA?

We now include this information in the Figure legend (plot is based on fluozin-1 experiment)